# Let Us Avoid a ‘Myopic View’ in Times of COVID-19

**DOI:** 10.3390/children9081125

**Published:** 2022-07-28

**Authors:** Lai Yien, Katherine Lun, Cheryl Ngo

**Affiliations:** Department of Ophthalmology, National University Hospital, Singapore 119074, Singapore; cfskl@nus.edu.sg (K.L.); cherylngo@gmail.com (C.N.)

**Keywords:** myopia, COVID-19, screen time, outdoor activity

## Abstract

The COVID-19 pandemic has affected people from almost all facets of life and it’s impact is extremely palpable among students. In this review paper, we discuss about the risk factors for myopia progression that were exacerbated by the pandemic, which are supported by evidence from studies published recently. It is imperative that measures are put in place to address the rising incidence of myopia so as to prevent the impending myopia pandemic.

## 1. Introduction

The World Health Organization (WHO) declared COVID-19 a pandemic on 11 March 2020, and the healthcare, economic, psychological, social, and environmental impact of COVID-19 can be witnessed in almost every country in the world. To combat COVID-19 and the increasing number of new cases at the onset, Singapore announced on 3 April 2020 a stringent set of preventive measures which were collectively called a ‘circuit breaker’. Similar lockdowns were also implemented worldwide, disrupting the lives of many through all age groups. According to the United Nations Educational, Scientific, and Cultural Organization, nearly 1.37 billion students (representing more than three out of four children and youth worldwide) from 138 countries globally were affected by these lockdown measures, with digital or e-learning approaches replacing face to face, classroom-based learning. This review paper aims to look at how lifestyle changes during COVID-19 may have precipitated the progression of myopia in school children and suggests preventive strategies to curb this increasing health burden.

## 2. Risk Factors for Developing Myopia

Epidemiological data have identified outdoor time as a key environmental determinant of myopia [1,2], and clinical trials have also found that increasing the amount of time that children spend outdoors is able to reduce the onset of myopia [3,4,5]. Furthermore, increased outdoor activities have other health benefits such as higher cardiorespiratory fitness [6]. A recent meta-analysis by Huang et al. found that more time spent on near-work activities was associated with higher odds of myopia, and that the odds of myopia increased by 2% for every one diopter-hour more of near work per week [7]. Diopter-hours were calculated using a cumulative near-work exposure variable according to the following formula to quantify near work exposure: 3 × (reading for pleasure hours + study hours) + 2 × (computer hours + video games hours) + watching television hours. In recent years, some studies have reported that shorter viewing distances for mobile phones and tablets may cause more significant myopia progression than projector and television screens due to the increased accommodation effort [8,9,10].

Moreover, screen time has other physical and psychological impacts on school children. For example, a recent study of 959 children in Japan found that those who reported longer screen time had greater body weight gain and had more behavioral problems such as irritability, inability to stop playing video games, frequent fights with siblings, excessive dependence on parents, and refusal to sleep independently [11]. In the United States, a national random sample of 2–17-year-olds also found that more hours of daily screen time were associated with lower psychological wellbeing, including less curiosity, lower self-control, more distractibility, more difficulty making friends, less emotional stability, being more difficult to care for, and inability to finish tasks [12].

## 3. COVID-19 in Singapore

In Singapore, the regulations to contain the spread of COVID-19 included a ban on private gatherings regardless of size with families and friends not living together, at home, or in public places such as parks, beaches, public housing void decks, and common property of private estates. Full home-based learning for school-going children was instituted in Singapore from 8 April 2020 to 1 June 2020 and from 27 September 2021 to 7 October 2021 as part of heightened safe distancing measures during the COVID-19 period. During this period, schools implemented video-conferencing platforms such as Zoom and Google Meet to engage students in live sessions, as well as lessons and homework to be completed on an online platform. Unfortunately, these necessary measures led to an increase in near-work activities, such as reading, writing, and screen time, and a decrease in outdoor activities, forming barriers to lifestyle modifications which have been shown to retard the progression of myopia.

Further aggravating the myopia epidemic situation, measures were implemented to postpone nonurgent clinic reviews to facilitate the deployment of adequate manpower and resources for the more pressing needs in dealing with COVID-19, as well as part of safe distancing measures to minimize local transmission. In the National University Hospital Department of Ophthalmology, appointments for children in the pediatric refractive clinic were deferred unless they had amblyopia, were at risk of developing amblyopia, or had any associated sight threatening conditions. Children who were on atropine eye drops to prevent the progression of myopia were given an extended prescription for the eye drop until their next appointment (approximately 9–12 months from their previous eye check). The caregivers of children who defaulted their appointments were contacted, and many expressed concerns of travelling to the hospital during this period. Both the deferment and the defaulting of appointments could also have contributed to progression of myopia, as there may have been a delay in the initiation of atropine eye drops or, for those who were already on atropine eye drops, a disruption in the timely monitoring and potential need for an increase in the concentration of atropine.

The Singapore Eye Research Institute’s Myopia Research Group is currently evaluating the data from Singaporean children, with the aim of contrasting the myopia progression levels before and during the pandemic.

## 4. Worldwide Impact of COVID-19 on Myopia

It is reasonable to hypothesize that the pandemic may precipitate the progression of myopia, as school closures have led to home-based learning using electronic devices while home confinement has reduced the time spent outdoors. To support this, there are increasing data from worldwide studies published on the worrying effect of the pandemic on myopia. 

### 4.1. Asia

Several studies were conducted in China to determine the effect of COVID-19 on myopia. In a study of 960 students in Shanxi Province, China, parents of these students were given a self-administered questionnaire which assessed the students’ lifestyle during the pandemic [13]. Results demonstrated an average screen time of 232 min per day compared with 114 min per day reported in a cross-sectional national survey conducted prior to the pandemic [14]. The same group reported a significant increase in the prevalence rate of myopia from 16.6% prior to the pandemic to 39.4% after the resumption of classes (with an interval of 6 months). 

Another group in Shanghai conducted a baseline examination for myopic children attending the hospital for regular follow-up visits, followed by two follow-up visits—visit 1 between October and November 2019, and visit 2 in May 2020 (after 4 months of lockdown in China during the pandemic) [15]. Results showed that the increase in myopia from visit 1 to visit 2 (−0.98 D) almost tripled compared with the baseline in visit 1 (−0.39). Similar to the previous study by Zhuang et al., a self-administered questionnaire was given to caregivers, and they reported nearly 10 times increased screen time during the pandemic (from 0.67 h per day at baseline to 5.24 h per day).

Liu et al. conducted a study via an anonymous online survey to 3405 school-age children (first to 12th grade) from 29 provinces and autonomous regions in China [16]. The questionnaire included questions to determine the children’s symptomatic changes in vision condition and the time spent using electronic devices and outdoor exercises, as well as indoor lighting conditions while using the electronic devices. They found that children who had myopia prior to the pandemic were more likely to report symptomatic myopia progression during COVID-19, and those who reported myopic symptoms engaged in an average of 3.7 more diopter hours than those who did not. In addition, results from the survey also showed that lighting conditions that were reported to be too dim or too bright were associated with progression of myopic symptoms. The same group reported in another study of a similar cohort (kindergarten to 12th grade) of 3831 children that every 1 h increase in daily digital screen use is associated with 1.26 OR higher risk of myopic progression [17]. They also found that the use of computers and smartphones was associated with higher risk of myopic progression that the use of television.

A recent publication by Yam et al. examined the myopia incidence and lifestyle changes of 1793 school children and found that there was a higher myopia incidence and increased spherical equivalent refraction and axial length progression in the COVID-19 cohort compared to the pre-COVID-19 cohort. During COVID-19, the 1 year incidence was 28%, 27%, and 26% for 6-, 7-, and 8-year-olds, compared to the 17%, 15%, and 15% respectively, before COVID-19. They also found that the time spent outdoors decreased significantly from 75 min/day pre COVID to 24 min/day during COVID-19, and the screen time increased from 2.5 h/day to almost 7 h/day [18]. Another prospective cross-sectional study by Wang et al. [19] found that a substantial myopic shift of −0.3 D was noted after home confinement for children aged 6–8 years. Hu et al. [20] reported a doubled myopic shift of spherical equivalent refraction and 7.9% higher myopia incidence in the post-COVID group compared with the pre-COVID group.

In India, complete lockdown started at the end of March 2020 and resulted in the confinement of school-going children from April to October 2020. Mohan et al. compared the rate of progression of myopia pre and during the COVID-19 pandemic in a study involving 133 children aged 6–18 [21]. The group found that 62.4% of children showed progression during the pandemic as compared with 45.9% prior to it. In the same study, 86.7% of children were reported to have ≥2 h per day of sun exposure pre COVID compared with 4.5% during COVID, and 2.3% had ≥2 h per day of mobile use for games pre COVID compared with 15.1% during COVID.

Although myopia has traditionally taken the limelight in Asian countries, children from Western countries are also not spared from the lifestyle changes caused by the lockdowns. 

### 4.2. Italy

A retrospective study conducted in Italy of 803 children aged 5–12 found a statistically significant decrease in the mean spherical equivalent following the COVID-19 lockdown, from 0.35  ±  1.75 D in 2019 to −0.08 ± 1.44 D in 2021 [22].

### 4.3. Spain

In Spain, home confinement and school closure were implemented in March 2020, where children were not allowed to leave their houses at all in the first 6 weeks. A cross-sectional study performed in September and October of 2020 including 1600 children aged 5–7 years old found that there was a decrease in the hyperopia rates and mean value of SE, along with an increase in the emmetropia rates, when compared with 2019 [23]. As could be expected, results from a self-administered questionnaire showed that, prior to confinement, 73% of these children spent >1.6 h per day outdoors and 60% spent >2 h per day doing near-distance activities pre confinement compared with 31% and 84% post confinement.

### 4.4. Turkey

A longitudinal study of myopia among the children aged 8–17 in Turkey demonstrated a significantly higher myopia progression of 0.71 D during the pandemic in 2020, compared with the values measured in 2018 (0.41 D) and 2019 (0.54 D). They also found that spending 2 h daily doing outdoor activities and living in a detached house were associated with less myopic progression [24].

## 5. Preventive Strategies

### 5.1. Reduce Screen Time

Teachers can ensure that there are adequate breaks in between video-conferencing and online work, as there is also increasing evidence to suggest that the intensity of near work, i.e., sustained reading at closer distance with fewer breaks, may be more important than the total hours of near work contributing to myopia progression [25,26,27]. In addition to measures in school, parents can also help to encourage good near-reading habits, such as reading in a room with adequate lighting and ensuring a good distance between the books or computer from the eyes, as well as being consistent in setting time limits to screen time for children. Indoor physical exercises, helping with household chores, arts and craft, simple baking/cooking, and audio books or free play may be helpful alternatives to screen time for children. Taking breaks from near work every 30–40 min may be beneficial and, while taking breaks, parents may also encourage children to look at distant objects or nearby greenery. In China, the Ministry of Education restricted the use of electronics as a teaching tool to no more than 30% of overall teaching time, <20 min per day spent on electronic homework (no more than 20 min), and prohibition of phones and tablets in classrooms [28]. Recently, China also introduced new regulations that limit the amount of time that children spend on video games to protect their mental and physical health.

In addition to the duration of screen time, there is increasing evidence showing that different types of electronic devices may have different effects on myopic progression, likely a result of the distance from the screen [8,9,10,17]. Hence, schoolteachers and parents can offer television screens or projectors over computers and smartphones as much as possible.

### 5.2. Increase Outdoor Activities

Sunlight reaches the retina at the back of the eye and triggers the production of a chemical known as dopamine [29]. Dopamine is an important neurotransmitter in the retina that mediates diverse functions including development, visual signaling, and refractive development [30]. Several studies have supported the hypothesis that light-stimulated dopamine antagonizes myopia development [31,32,33,34,35]. Hence, children who do not spend enough time outside are not exposed enough to light from the sun, whereby dopamine cannot be released in the retina, which in turn reduces the stop signal for homeostatic control of myopic eye growth [29]. A study in Singapore found that Singaporean children spent only 1–1.5 h outdoors each weekday, and 1–2 h outdoors per day on weekends [36]. In Taiwan, a myopia prevention program, the ‘Tian Tian 120 outdoor program’, was implemented where schools are encouraged to take students outdoors for 120 min/day. They found that the long-term trend of increasing prevalence of reduced visual acuity in school children from 2001 to 2011 (34.8–50%) was reversed from 2012–2015 (49.4–46.1%) after the program was implemented [4]. This highlights the positive and far-reaching effects that public policies can have in reducing myopia progression and is certainly a model that other countries can emulate. 

### 5.3. More Effective Methods to Track Outdoor and Near-Work Exposure

The study by Liu et al. showed that lighting conditions were associated with symptoms of myopic progression [16]. However, most of these behavioral investigations were dependent on self-reporting [16,21,23], which may have introduced response bias. Ideally, a more objective method should be used for these measurements. For example, there have been several devices introduced recently to track exposure to bright light such as Actiwatch [37], Fitsight [38], HOBO [39], and Clouclip [40]. The ability to reliably review children’s outdoor exposure and lighting condition may then allow eye care professionals to better counsel and advise on lifestyle modification. 

## 6. Conclusions

The prevalence of myopia has increased markedly in the past decades such that 80–90% of 18-years-olds in East Asia are currently myopic and 10–20% are currently highly myopic [28]. Although most people with myopia will not experience sight-threatening complications, high myopia is associated with retinal detachment, choroidal neovascularization, and glaucoma which can lead to irreversible vision loss. As such, while we focus on the measures to reduce the transmission of COVID-19, it is prudent to keep a macroscopic view to identify and prevent potential consequences that can arise from these measures. Even as we are making progress in the battle against the COVID-19 pandemic, and the world is starting to open up again, it is important not to lose sight of the impending ‘myopia pandemic’. School- and community-based programs are important in encouraging and increasing time spent outdoors daily. Positive parent attitudes on after-school outdoor activities and support for educators on outdoor learning initiatives for school children are crucial.

## Data Availability

Not applicable.

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
