# Peer review of "Let Us Avoid a ‘Myopic View’ in Times of COVID-19"

_children, 2022, doi:10.3390/children9081125_

Round 1

Reviewer 1 Report

This is a well-written review about myopia progression during COVID-19 pandemic.

Just a few elaborations are helpful to readers who is not an ophthalmologist.  

Could you give an detailed explanation of  'SER' (page 2 line 54) and 'one diopter-hour more of near work' (page 2 line 67)? 

Author Response

Thank you for your kind review.

  1. Apologies for using the abbreviation, have spelt out SER in full (spherical equivalent refraction.
  2. Diopter-hours were calculated using a cumulative near work exposure variable using the formula: 3 x (reading for pleasure hours + study hours + 2 x (computer hours + video games hours) + watching television hours to quantify near work exposure. 

Reviewer 2 Report

Brief summary

The current review discussed the factors which may influence myopia progression during the COVID-19 period, this topic is current popular and fascinating. However, as a review, the current manuscripts lose the basic structure, such as abstract, research gap, and critical thinking. Besides, most of the references discussed the risk factors in myopia, while most of them were irrelevant to COVID-19. The authors did determine how the pandemic influenced myopia progression. Therefore, the current manuscript is not an academic review article.

Specific comments:

Page1 line 6 to 42. The authors described the background of COVID-19 and related policy in Singapore, while there was no research gap and why it is important to do a review. The main part of the review started from line 43: “there are increasing data from…on myopia.”

From line 62 to 119, although the authors introduced risk factors in myopia, such as less outdoor activities,  near work, and screen time, it seems it directly extracted from the “review of risk factors in myopia”. The authors need to provide more references about how the lifestyle changed during COVID-19 and the link between the pandemic with more myopia progression.

As a review of COVID-19,  there were only 5 papers related to the pandemic in the current manuscript, which did not cover the completeness of the topic. Thus, the reviewer suggested the authors could combine more related COVID-19 papers published recently.

Line 116 to 128, similar to my previous comments. This paragraph could be presented in any “risk factor review”. As the risk factors and school- community-based program for controlling myopia has already been investigated and conducted before COVID-19. Hope the authors could add more specific argument related to the pandemic.

Author Response

Dear reviewer,

Thank you so much for your insightful input to improve on our paper. We have included more papers performed pre and post-COVID looking at the impact of COVID-19 on lifestyle changes and how that may have affected myopia progression and look forward to having your comments and feedback again 

Round 2

Reviewer 2 Report

The current manuscript has been greatly improved, and the authors have solved most of the comments. Some minor comments are in the following:

Line 109 please add the unit: visit 2 (-0.98 D) almost tripled compared with baseline to visit 1 (-0.39 D).

Line 174: should be: 2 hours daily

Line 211: “A study published in April last year in the British Journal Of Ophthalmology, found that Singaporean children spent only one to 1½ hours outdoors each weekday, and one to two hours outdoors per day on weekends.” It is not necessary to mention the Journal name, and “ published in last year” is unclear. So the authors could change to: “ A study in Singapore spend only 1 to 1.5 hours outdoors each weekday, and 1 to 2 hours outdoors per day on weekends.

Line 213: “ This is lower than the time children in Australia and Britain spent outdoors - an average of two to three hours outdoors each day” please add the reference.

Line 225: “The study by Liu et al showed that lighting conditions were associated with symptoms of myopic progression.” add the reference. “One limitation across the studies quoted in this paper is the dependence on self-reporting …… near work exposure and lighting condition.” It is odd that the authors talked about the limitations of this references suddenly. As ref 23 (Alvarez-Peregrina C et al.,), ref 21 (Mohan et al.) also used self-report methods.

Line 225 to 228: this paragraph, “The study by Liu et al showed that lighting conditions were associated with ……near work exposure and lighting condition.” It could be changed as: “However, most of these behavioral investigations (ref 16, 21,23) dependence on self-reporting. Ideally, a more objective method should be used for these measurements. For example, …… counsel and advise on lifestyle modification.”
